# A Novel Path Planning Strategy for a Cleaning Audit Robot Using Geometrical Features and Swarm Algorithms

**DOI:** 10.3390/s22145317

**Published:** 2022-07-16

**Authors:** Thejus Pathmakumar, M. A. Viraj J. Muthugala, S. M. Bhagya P. Samarakoon, Braulio Félix Gómez, Mohan Rajesh Elara

**Affiliations:** Engineering Product Development, Singapore University of Technology and Design, Singapore 487372, Singapore; pathmakumar_thejus@mymail.sutd.edu.sg (T.P.); bhagya_samarakoon@mymail.sutd.edu.sg (S.M.B.P.S.); braulio_felix@mymail.sutd.edu.sg (B.F.G.); rajeshelara@sutd.edu.sg (M.R.E.)

**Keywords:** audit robot, geometrical feature, cleaning auditing, swarm algorithms

## Abstract

Robot-aided cleaning auditing is pioneering research that uses autonomous robots to assess a region’s cleanliness level by analyzing the dirt samples collected from various locations. Since the dirt sample gathering process is more challenging, adapting a coverage planning strategy from a similar domain for cleaning is non-viable. Alternatively, a path planning approach to gathering dirt samples selectively at locations with a high likelihood of dirt accumulation is more feasible. This work presents a first-of-its-kind dirt sample gathering strategy for the cleaning auditing robots by combining the geometrical feature extraction and swarm algorithms. This combined approach generates an efficient optimal path covering all the identified dirt locations for efficient cleaning auditing. Besides being the foundational effort for cleaning audit, a path planning approach considering the geometric signatures that contribute to the dirt accumulation of a region has not been device so far. The proposed approach is validated systematically through experiment trials. The geometrical feature extraction-based dirt location identification method successfully identified dirt accumulated locations in our post-cleaning analysis as part of the experiment trials. The path generation strategies are validated in a real-world environment using an in-house developed cleaning auditing robot BELUGA. From the experiments conducted, the ant colony optimization algorithm generated the best cleaning auditing path with less travel distance, exploration time, and energy usage.

## 1. Introduction

Cleanliness is one of the inevitable factors that span from an individual’s living space to the growth index of developing and developed nations. The professional cleaning industry is a steeply growing field, valued at over $46 Billion U.S Dollars in 2020 forecast to grow 10% by the end of 2026 [1]. Amidst coping with pandemics like COVID-19, the demand for cleaning services has increased steeply in recent years [2]. For the past decade, we can observe a successful usage of leading-edge technologies, including robots for efficient cleaning in both domestic and professional cleaning domains [3,4,5,6]. The necessity for high-quality performance, efficiency, and favorable factors from the industry has paved the way for the successful application of robotics technology in the field of automated cleaning.

A vast volume of research has been reported on enhancing the quality of robot-aided cleaning for the past five years. Most reported research focuses on complete coverage planning, energy-aware cleaning, multi-robot cooperation, etc. For instance, a scalable approach for full coverage planning for cleaning robots has been reported [7]. Furthermore, a sector-based online complete coverage planning to bridge the shortcomings of cleaning robots with limited sensing and computation resource is proposed by Lee et al. [8]. The problem of ensuring cleaning efficiency is addressed from a different perspective by introducing optimal footprint for robots alongside the conventional complete coverage path planning approaches [9]. The work mentioned above is validated on a vertical surface cleaning robot by performing an effective hydro-blasting for ship-hull cleaning.

Noh et al. [10] presented an energy-aware cleaning path to enhance the cleaning efficiency using deep reinforcement learning-based approach for energy-aware cleaning. Besides the coverage planning, multi-robot collaboration is another area widely utilized for improving the cleaning quality. The work mentioned in [11] proposed an online heterogeneous multi-robot collaboration system for cleaning robots. The work discussed above highlights a scalable approach for robots with limited sensing capabilities to maximize cleaning performance. The concept of a multi-robot system for cleaning is also explored in the case of oil spill cleaning [12]. The system mentioned above proposes an aerial multi-robot system for an optimal strategy for the contaminated area with minimal wastage of dispersants. The efforts towards improving automated cleaning are centered on improving the efficiency of cleaning robots or machines rather than analyzing the cleaning quality it delivers. Even though the technical advancements in the field of automated cleaning are significant, the analysis of cleaning quality provided by the cleaning robots remains as naive as manual inspection.

Auditing the cleanliness of a region is an important factor to be considered in every cleaning and maintenance task. The attempts to audit the cleanliness are reported in the field of food processing industries and hospitals. For example, the work presented by Giske et al. [13] explores a comparative study between the quality of cleaning delivered by robots and manual cleaning methods. As mentioned earlier, the research discusses validating the effectiveness of cleaning using micro-biological analysis. An AdenosineTriphosphate (ATP) bioluminescence technique is another method used to assess the quality of cleaning by estimating the microbial colony presence on the surface of interest. Lewis et al. presented the usage of ATP bioluminescence to benchmark the quality of cleanliness [14]. In the work mentioned above, the authors benchmarked the quality of cleanliness for hospitals in relative light units (RLU). Similarly, Asgharian et al. presented the systematic procedures and guidelines for cleaning quality analysis in the pharmaceutical industry [15].

Similarly, a cleaning assessment report generation based on surface swabbing followed by a laboratory analysis as detailed in [16]. Even though the cleaning quality assessment is regarded as an essential practice in every domain, effort towards establishing a quality assessment is reported in a few areas and is limited to hospitals, pharmaceuticals, and food processing industries [17,18]. Besides, the current methods for micro-biological analysis are laborious and not scalable for cleaning auditing for built infrastructure and domestic cleaning. The primary effort to establish a scalable cleaning auditing is reported in [19] which is the robot-aided cleaning auditing framework using a sample auditing sensor. The sample audit sensor performs adhesive dust-lifting followed by visual analysis to provide a sample audit score for a sample region of 2 cm × 2 cm. The framework uses repeated sample auditing for a vast area using an audit robot that carries the sample auditing sensor as the functional payload. Using the approach mentioned above, the levels of cleanliness for a region are estimated by combining the result of all the sample audits performed in the area of operation. Besides computing the overall auditing score for a region, the work mentioned above provides an empirical estimation of the quality of cleaning. The autonomous robot-aided cleaning auditing can be a potential solution for performing cleanliness inspection effectively without human interventions and laboratory-oriented procedures compared to the existing cleaning quality analysis. A significant challenge in the development of robot-aided cleaning auditing is the formulation of an efficient exploration strategy for inspection. The work mentioned in [19] used a frontier exploration strategy for the robot to explore its area of operation and periodic pattern to determine the region for auditing. However, in work mentioned above, frontier exploration does not guarantee the robot to explore dirt-accumulated areas. The shortcomings of the frontier-exploration-based auditing strategy and sampling decision are improved using reinforcement learning-based exploration [20]. In the research mentioned above, the audit robot uses its experience learned from the modeled environment for exploration and making sampling decisions. This work presents a first-of-its-kind path planning approach dedicated to cleaning auditing robots. The autonomous mobile robots that perform similar tasks to the auditing, especially floor cleaning and ground patrolling robots, use the complete coverage planning strategy using an optimization technique to yield the best path [21,22]. However, adopting a similar complete coverage planning for audit robots is not effective because:for a large environment, collecting samples uniformly across the complete region are not time-efficientthe dirt accumulation pattern is chaotic and often clustered in regions left unattended by cleaning robots.lack of an effective method that determines from where to gather dirt samples for an effective cleaning auditing.

This research work bridges the challenges mentioned above by proposing a novel path planning strategy that is driven by geometric signatures of the environment for cleaning auditing robots. To the best of the author’s knowledge, a path-planning approach considering geometrical signatures of the environment has not been explored in the domain of cleaning robots or cleaning audit robots so far. Due to less accessibility and the inherent navigation behavior of cleaning robots to avoid obstacles, the walls and corners often become the site for possible dirt accumulation compared to the other regions for a regularly cleaned area. Therefore, a geometrical feature extraction framework is devised to identify the probable dirt accumulation region from a 2D map. Swarm algorithms are exploited to plan an efficient way to cover the sample locations.

The general objective of this research work is subdivided into three:formulate the dirt location identification method by extracting the geometrical signatures from the environmentdevise an efficient path that connects all the identified dirt location (i.e., sample locations) by minimizing the energy costvalidate the geometry-based dirt location identification in a real environment after repeated cleaning trialsexperimentally validate the optimal path generated by the proposed method in a real environment using an in-house developed cleaning audit robot

The rest of the article is structured as the Section 2 provides the background study conducted for devising the proposed approach. This is followed by Section 3 that presents a bird’s eye view of the proposed approach. The detailed description of path generation strategy is provided in Section 4 and Section 5. The Section 6 provides a detailed description of the conducted validation trials, followed by the Section 7.

## 2. Related Works

Two significant aspects of the proposed path planning strategy are (1) identification of probable dirt locations in a 2D map using geometrical features and (2) generating an optimal path through probable dirt locations using swarm algorithms.

### 2.1. Geometrical Feature Extraction

Feature extraction and description is a key element in perception for autonomous robots. Moreover, a geometrical feature like line segments, curvature, and corners is used over the conventional feature extraction techniques like SIFT [23], ORB [24], AKAZE, [25] etc. A SLAM algorithm that uses descriptors for line segments is reported in [26]. Similarly, the curvature property of road lane is used as a key feature for pose estimation for autonomous vehicles is reported in [27]. Yan et al. used corner features for two-dimensional SLAM [28]. Besides, the geometrical feature extraction methods are also widely used for hand gesture recognition [29], finger-knuckle-print verification [30], etc. Among the applications of geometrical feature extraction, the classical method for the extraction of boundaries in a 2D matrix is canny edge detection [31]. Another popular approach is estimating the boundaries based on Hough transformation for the line geometry detection [32,33,34]. The popular approach for corner detection is the Harris Corner detector [35]. Similarly, a robust Anisotropic Directional Derivative (ANDD) filter-based corner detection is another classic method for corner extraction [36]. Chord-to-Point Distance Accumulation (CPDA) is another method reported to have low localization error and high repeatability [37]. Similarly, machine learning-based detectors have known for accuracy and repeatability in detecting corners from a given 2D image [38,39].

### 2.2. Optimal Path Generation

Swarm algorithms and evolutionary algorithms are widely adopted for solving the optimal path planning algorithms with multiple constraints [40,41,42,43,44]. Whale optimization (WO) based path planning for an underwater robot is mentioned in [45], where the optimization techniques are used for generating a path with safe and minimal fuel consumption. An improved sparrow search algorithm for path-planning for a mobile robot is reported in [46]. Furthermore, hybrid Quantum-behaved Particle Swarm Optimization (HQPSO), a variant of classical PSO, has been used for path generation for an unmanned underwater vehicle (UUV) [47]. Modified ant-colony optimization (ACO) is used for path planning for AGV-based parking system is detailed in [48].

## 3. System Overview

### 3.1. Cleaning Auditing Overview

The robot-aided cleaning auditing is a three-step procedure detailed in [19]. For completeness, we briefly explain the process of the robot-aided cleaning auditing framework. Figure 1d shows the overview of robot-aided cleaning auditing. The first step in robot-aided cleaning auditing is called sample auditing, where the cleanliness of a sample area is determined. The second step is space auditing, where repeated sample auditing for a larger area is performed. The space auditing is facilitated by an in-house developed audit robot called BELUGA, facilitated by exploration algorithms to achieve efficient sample auditing in different locations. The BELUGA robot is a differential drive mobile robot equipped with sensors for navigation and perception (shown in Figure 1c). The robot maps a given area and does localization within the map using Adaptive Monte Carlo Localization (AMCL) method. The perception, localization, and control algorithm are executed using an onboard embedded computer. The key payload carried by the BELUGA robot is the sample auditing sensor (shown in Figure 1a). The sensor consists of adhesive tape for dust-lifting and a digital microscope for analyzing the adhesive tape surface. The field of view of the digital microscope has an active light source to maintain constant ambient light throughout the operation such that variation in light intensity does not affect the sensor operation. Figure 1b shows the dust extracted by the sample audit sensor viewed by its built-in digital microscope. The sensor uses the structural similarity index(SSIM) of the tape surface images before and after dust-lifting to estimate the magnitude of dirtiness of the surface [19]. The sensor reports the magnitude of cleanliness as a sample audit score to evaluate the overall cleanliness of the sample region (sample auditing). The comprehensive cleanliness estimation is done by conducting repeated sample auditing for a larger area with the help of the BELUGA robot (space auditing). With the exploration strategies, the robot goes to specific locations to perform sample auditing.

### 3.2. Path Generation Overview

A bird’s eye view of the proposed path planning framework is detailed in this section. Figure 2 shows the overall architecture of the proposed path generation framework. The process pipeline has three components—geometry extraction, sample selection, and path generation. The input to the system is the 2D occupancy grid of the environment, which is generated by the simultaneous localization and mapping (SLAM), a.k.a mapping method. From the 2D occupancy grid, the locations for sampling are selected based on the environment’s geometrical signatures that contribute to the dirt accumulation. The geometry extraction and sample selection procedures achieve the process mentioned above for dirt location identification. The geometry extraction procedure comprises boundary extraction, corner extraction, and free-space extraction from the given occupancy grid. The sample selection procedure selects the sample locations in the nearest proximity to boundary maps and corner maps. The general sample points are a set of locations in the occupancy grid obtained by uniform grid sampling. The sample locations are identified by combining the boundary samples, corner samples, and random samples. The random samples are a few random locations in the free space. A detailed information regarding the dirt location identification strategy mentioned above is presented in Section 4.

Once the sample locations are identified, the robot can visit the locations in any order to do sampling and auditing. However, choosing the optimal way to visit the locations and perform the sample auditing is important. The optimal path generator in the proposed framework takes in the locations identified after the sample selection procedure and generates an efficient path with minimum time and energy. With the help of navigation algorithms, [49], the robot follows the path and performs auditing for a given area.

## 4. Dirt Location Identification

This section presents the identification of probable dirt locations (sample audit locations) from a given occupancy grid. Considering the natural tendency of dirt accumulation nearer to the corners and wall, the probable dirt locations are identified by selecting sample points in close proximity to walls and corners. To identify the dirt locations for the robot to perform sample auditing, the entire occupancy grid is evenly sampled with a resolution proportional to the robot footprint and locations corresponding to the samples are regarded as general sample locations. We defined the general sample locations as the set of locations that are equally spaced in the free-space region in an occupancy grid Figure 3c. While generating the general sample locations, the occupied cells and unknown cells in the grid are discarded. The sample audit locations are a subset of the general sample location that lies closer to the corners and boundaries in the occupancy grid. Hence, the main task involved in sample location identification or the probable dirt location is to extract the corners and boundaries from the occupancy grid. Since the occupancy grid a.k.a map is represented as a two-dimensional 8-bit matrix, where each element represent the three states of occupancy (occupied, free and unknown). Hence the conventional image processing algorithms can be applied over an occupancy grid. Figure 3a shows the representation of an occupancy grid.

### 4.1. Boundary Sample Extraction

The Figure 3d illustrates the critical operations involved in the boundary extraction procedure. For boundary extraction, we performed the following methods:General sample location identification: The general sample locations are obtained by sampling the free space in a uniform fashion. The general sample location $P^{general}$ is obtained by grid sampling the occupancy grid $S_{i,j}$ in intervals Δu and Δv along rows and columns.Occupancy grid thresholding: Each cell in an occupancy grid shows the probability of occupancy. A two state binary map *B* has been generated by applying a thresolding such that B(i,j)=1 if $S\left(i,j\right)<T_{max}$ else B(i,j)=0, where S(i,j) is the occupancy grid and $T_{max}$ is the maximum occupancy threshold.Boundary extraction: On the binary map, we used Zhang-Suen thinning algorithm [50] for extracting the contours. However, the contour extraction results in detecting all closed contours in the occupancy grid, including the undesirable contours that form over the obstacles. The largest contour $C_{large}$ from all the set of contours $C_{i}$ is regarded as the boundary region.Selection of boundary location: The contour corresponding to the boundary region $C_{large}$ is sampled in a regular interval to obtain a set of points $B_{i,j}$ that lies on the contour. The boundary locations $L^{boundary}$ are selected from the general sample S(i,j) points that lie in the distance less than $R_{b}$.

The location $L^{boundary}$ is the sample location that lies closer to the walls such that performing sample auditing at $L^{boundary}$ will result in analyzing dirt accumulations contributed by the walls in a region. The distance $R_{b}$ decides how many samples closer to the walls have to be considered for sample auditing.

### 4.2. Corner Sample Extraction

The corner Sample extraction follows a similar approach to boundary extraction.

The general sample location $P^{general}$ is identified by performing uniform sampling of occupancy grid $S_{i,j}$.Corner extraction: The machine learning-based fast corner extraction algorithm [39] is used for identifying the corner locations.Selection of corner location: Similar to the boundary location identification, the corner locations $L^{corner}$ are selected from the general sample S(i,j) points that lie in the distance less than $R_{c}$.

The distance $R_{c}$ decides how many samples closer to the sharp corners have to be considered for sample auditing.

#### Random Samples

Few random locations $L^{random}$ are selected from the general sample location $P^{general}$ for spanning auditing to the complete area. However, the random locations are smaller in number, and it is selected based on the size of the occupancy grid. The locations for auditing are a combination of corner locations, boundary locations, and random locations. The Equation (Equation 1) represented the set of locations for auditing.
(1)$$S=L^{corner}\cup L^{boundary}\cup L^{random}$$
where $L^{corner}$, $L^{boundary}$, and $L^{random}$ represents corner locations, boundary locations and random locations respectively.

## 5. Optimal Path Planing

After determining the probable dirt locations, the robot has to visit each location once and perform the auditing. Here, the determined probable dirt locations ($P_{i}$) are considered as the waypoints of the robot’s navigation path. The robot is assumed to be initialized at a designated starting location in the workspace (denoted as $P_{1}$). Two example scenarios are depicted in Figure 4, where there is *N* number of locations to be visited, including the starting point. Similarly, there can be many sequences of waypoints where the robot can visit all the locations at one time.

The robot must determine an optimum sequence of these waypoints that minimizes the energy usage for an efficient auditing process, as the energy-efficient coverage is vital for a robot deployed in maintenance and inspection domains [51,52]. The optimum sequence of waypoints is defined as $\left\{W_{k}\right\}$ such that k=1 to N, where *N* is the number of waypoints, including the starting position. The energy used by the robot for navigation is proportional to the distance traveled by the robot in the event of a level workspace. The energy usage of navigating from kth waypoint to lth waypoint, $E_{k,l}$ can be estimated as (Equation 2) where $D(W_{i},W_{j})$ represents the navigation distance between kth and lth waypoints, and *M* is a proportional constant. The navigation distance $D(W_{i},W_{j})$ is estimated considering the *A*∗ algorithm for a collision-free path between $W_{i}$ to $W_{j}$.
(2)$$E_{k,l}=MD\left(W_{k},W_{l}\right)\;\;\mathrm{where}\;k\neq l$$

The total energy usage of the robot, *E*, for accomplishing the navigation for auditing can be formulated as in (Equation 3).
(3)$$E=\sum_{k=1}^{N-1}E_{k,k+1}$$

The energy usage of the robot for navigation between all the pairs of the waypoints could be estimated, and the total energy requirement could be estimated by considering all the possible sequences to find the sequence that results in the lowest energy usage. However, the number of possible pairs of waypoints becomes N(N−1)/2. There exist (N−1)! combinations for joining these waypoint pairs. Thus, determining the path of lowest energy usage through evaluating the energy usage for all the possible paths becomes inefficient and complex when *N* is high. Moreover, this problem is a no polynomial-time known solution problem (NP hard problem).

Swarm optimization algorithms are effective techniques for finding a proper solution for this kind of problem [53,54]. Two versatile swarm algorithms, Ant Colony Optimization (ACO) [55] and Particle Swarm Optimization (PSO) [56] are used here to find the optimal sequence of waypoints considering the minimization of the cost function given in (Equation 3). These two optimization techniques are selected since they are well known for the convergence to the global optima in similar problems.

The path generation problem considered here is analogous to the classical Travelling Salesman Problem (TSP). However, in most of the off the shelf tools for classical TSP, it is assumed that there are no obstacles in between the nodes and hence Euclidean distances between the nodes are considered for the optimization. In contrast, for the cleaning auditing, the robot has to operate in obstacle-cluttered environments where the robot must find a collision-free path between two nodes. The *A*∗ algorithm is used to find the collision-free path between two nodes. The distance of the *A*∗ path is used as the distance between two nodes.

### 5.1. Particle Swarm Optimization (PSO)

Particle Swarm Optimization (PSO) is inspired by the social behavior of birds flock or fish school. The cooperation of individuals in a swarm based on individual and group knowledge toward finding a goal is utilized in this technique. The flow of the PSO algorithm is given in Figure 5. Here, each individual is considered as a particle with a position and a moving velocity. Each particle is randomly initialized with a velocity and a position at the start. Then, the algorithm iteratively attempts to find the optimal solution. The fitness of each particle is evaluated for the current solution, and the global and the local best positions are updated as per the evaluated finesses. Then, the new velocity and position of each particle in the swarm are calculated. This process is repeated until a stopping criterion is met. The global best at the time of stopping the iteration is the final optimum solution.

The parameters of the PSO algorithm have been set as follows in this work by observing the performance variation. The population size was chosen as 100. The inertial weight and the inertial weight damping ratio were configured to 0.9 and 0.95, respectively. Global and local learning coefficients were set to 0.85. Reaching 1000 iterations was defined as the stopping criterion.

### 5.2. Ant Colony Optimization (ACO)

Ant Colony Optimization (ACO) is inspired by the foraging behavior of some ant species. The ants lay pheromone on the ground to mark direct peer ants toward resources such as food sources while exploring the habitat. The flow of the ACO algorithm is depicted in Figure 6. At the start, ants and the pheromone trails are initialized. Each ant represents a solution. Then, the paths found by ants are compared. In other words, the fitness value of each ant is evaluated. Subsequently, the pheromones are updated based on their fitness levels. This process is iterated until a stopping criterion is met. The best solution at the termination of the algorithm is the optimal solution.

The parameters of the ACO algorithm have been configured as follows based on performance observations. The number of ants was set to 100. The evaporation coefficient was set to 0.15. The effect of ant sight and traces were chosen as 1 and 4, respectively. Reaching 1000 iterations is defined as the stopping criterion.

## 6. Results and Discussion

We have carried out multiple experiment trials to validate the proposed approach. We conducted two sets of experiments to analyze the performance of the proposed approach. The first set of experiments quantifies the performance of dirt sampling with proposed geometrical feature-based dirt location identification in real-time. The second set of experiments analyzes the behavior of the path generated by the proposed framework in different environments.

### 6.1. Trial-1: Sample Location Identification

The performance of sampling with geometrical features based on dirt location is analyzed by defining the dirt gathering efficiency, which is the ratio of the number of dirt samples collected to the total samples collected as given in Equation (Equation 4),
(4)$$\eta_{dirt}=\frac{N_{d}}{N_{d}+N_{c}}$$
where $N_{d}$ and $N_{c}$ represent the number of dirt samples collected and the number of clean samples collected. The factors $N_{d}$ and $N_{c}$ are determined by counting the number of dirt particles gathered using dirt lifting followed by computer vision-based dirt counting. The dirt specks on the adhesive tape are treated as the connected pixels on the images (also known as blobs) captured by the microscope. Steps involved in computer vision-based dirt counting include:Apply thresholds on the source image and convert the image to binaryUsing contour extraction, identify the connected pixels from the binary image and estimate blob centroidThe blob centroid is regarded as the location of the dirt particleThe number of centroids is regarded as the dirt count

The blob detection algorithm is implemented using OpenCV libraries given in [57,58].

The Figure 7 shows the dirt speck identified and counted using blob detection. Our first set of the experiment consists of two trials, trial-1.a and trial-1.b, in which the trials replicates a cleaning routine carried out using a commercial cleaning robot that does a zig-zag path planning. For validating the probable dirt location identification, we created a sample environment of dimensions 4.5 m × 4.5 m. The domestic floor cleaning robot with autonomous navigation capabilities is operated for 15 min to replicate a cleaning routine. To analyze the dirt accumulation, dirt particles are uniformly distributed (fine ground tea powder with a particle size 0.5 mm–1 mm) before the operation of the cleaning robot. The cleaning routine is repeated for five rounds. After five rounds of cleaning operation, the dirt samples are gathered based on uniform sampling and at the location candidates obtained from the proposed geometrical features.

The comparison of dirt particle counts at locations provided by the algorithm with the locations selected by uniform sampling is recorded. The trial-1.a and trial-1.b differ in terms of the obstacle density. Figure 8a shows the sample locations identified for the trial-1.a. Figure 8b shows the locations identified by the proposed geometrical feature-based dirt location identification.

Figure 8c shows the dirt counts recorded for uniform sample gathering and Figure 8d shows dirt count obtained from the proposed approach. Similarly, Figure 9a shows the sample locations identified for the trial-1.b and Figure 9b shows the locations identified by the proposed method. Figure 9c shows the dirt counts recorded for uniform sample gathering and Figure 8d shows the dirt count obtained from the proposed approach. In both trials, a dirt count equal to 10 is regarded as the threshold for classifying the gathered sample as dirty or clean. This implies that all gathered samples having a count above 10 is regarded as dirt sample. The dirt gathering efficiency (Equation (Equation 4)) is computed for trial-1.a and trial-1.b. The result of dirt gathering efficiency calculation for trial-1.a and trial-1.b is tabulated in Table 1 and Table 2, respectively.

In trial-1.a, the sample locations identified using the geometrical signatures extracted from the maps were more concentrated toward the walls and corners. In trial-1.a, 51 locations are identified for using the proposed approach. The locations closer to the wall and corners had more relatively dirt counts than other locations. It is also observed that few locations near the walls, the robot cleaned well and left fewer dirt counts. However, the general pattern observed is the dense accumulation of dirt nearer to the location identified by the proposed approach. The dirt gathering efficiency of 0.92 for the proposed approach confirms the above-mentioned observations. In trial-1.a and trial-1.b, we have observed sample locations are not identified by the proposed approach in certain regions around the walls. This is because of the imperfections in the LiDAR scan while generating the map. In trial-1.b, the sample locations identified using the geometrical signatures extracted from the maps were also more concentrated towards the walls and corners and sparse dirt accumulation in the central locations. In trial-1.b, 59 locations are identified using the proposed approach. The proposed method showed more number of locations for trial-1.b because of more number of walls and corners introduced by additional obstacles in the environment. Similar to the previous trial, with few locations near the walls, the robot cleaned well, leaving fewer dirt counts. However, the dirt accumulation was dense near the location identified by the proposed approach. A dirt gathering efficiency of 0.74 is recorded for the proposed approach, and 0.54 is recorded for the uniform sampling method. We observed the robot took more turns near the region bounded between two obstacles and walls, resulting in multiple passes through the same location. This resulted in a drop in dirt gathering efficiency for the proposed approach. However, there is a significant improvement in dirt gathering efficiency in both experiment trials.

### 6.2. Trial-2: Path Generation

Our second set of experiment trials validates our proposed path generation framework. Trial-2 consists of experiments conducted in three real-world environments with the BELUGA robot. Each environment is different regarding the area of operation for cleaning auditing. The first environment (environment-1) is an indoor lab space with approximately 58 m${}^{2}$ of the total area accessible. The obstacle density in environment-1 is higher; however, the obstacles are placed in a well-ordered manner. The environment-2 is a semi-indoor pantry area. The obstacle density in environment-2 is slightly more than in environment-1. The third environment, environment-3 is a ramp entrance where the environment is more complex in terms of shape and orientation of obstacles. The Figure 10 shows the operation of the BELUGA robot in all three environments. Four sets of experiment trials are conducted in each environment considering PSO, ACO, Zig-Zag, and random path planning. The test environments considered for the experiments have a moderate number of points to visit. However, we are targeting the cleaning audit in the large environment such as shopping malls. In that case, the number of points will be much higher, and swarm-based optimization methods would be more suitable than the other approaches. Zig-zag path planning is one of the common path planning methods used in the domain of cleaning robotics. Hence, it would be worthy of considering zig-zag path planning as a baseline for comparison in cleaning auditing applications along with random path selection. Here, Zig-zag path planning is considered along the *Y*-axis where the robot starts the selection of the point which it has the least coordinate in *Y*-axis. Then, the point with the second least Y coordinate is selected as the second point. This ordering pattern is continued until all the points are selected.

In every experiment trial, the total path covered by the robot, the total time taken for completing the sampling, and the current consumption from the robot are recorded. The total energy taken for the exploration is computed using the Equation (Equation 5):(5)$$E=\int_{0}^{T}v\left(t\right)i\left(t\right)\;dt$$
where v(t) is the terminal voltage of the battery, *T* is the total exploration time, and i(t) is the instantaneous current reading from the battery management system of the robot. The overall observations recorded in trial-2 are tabulated in Table 3. The convergence results of the PSO and ACO algorithms are given in Figure 11.

In environment-1, the algorithm has identified 29 sample points. The experiment trials in environment-1 are represented as trial-2.a, trial-2.b, trial-2.c and trial-2.d. In trial-2.a, the robot did sample auditing in the identified locations by selecting the points randomly and following an A∗ path connecting the selected points (shown in Figure 12a). In trial-2.b, the robot did sample auditing in the identified locations by selecting the points in a zig-zag fashion along the *Y*-axis. Similar to the previous trial, the robot followed an A∗ path connecting the selected points (shown in Figure 12b). In trial-2.c the point selection is made based on the PSO algorithm; in trial-2.d, the point selection is made based on the ACO algorithm. From trial-2.a to trial-2.d, we could observe that the optimal path generated by the ACO algorithm has a shorter path length than all other methods. The most sub-optimal strategy was a random selection of points. The robot took 427 s to complete the path generation in the case of the ACO algorithm with a total energy consumption of 22.55 kJ (Kilo-Joule). From the observations made from environment-1, it is evident that ACO shows the best convergence in terms of optimizing the path length and energy consumption. The convergence results shown in Figure 11a,b also verify the proper convergence of the ACO and PSO algorithms in this case.

In environment-2, the algorithm has identified 41 sample points from 93 m${}^{2}$ of area. However, the robot could access only 39 points and 2 points were too close to the obstacle and omitted during navigation.

Four sets of experiment trials are conducted in environment-2, represented as trial-2.e, trial-2.f, trial-2.g, and trial-2.h, respectively. Similar to environment-1, the robot did sample auditing by selecting the sample points randomly and following an A∗ path connecting the selected points in trial-2.e (as shown in Figure 13a). Similarly, the robot did sample auditing in the identified locations by selecting the points in a zig-zag fashion along the y-axis in trial-2.f. The path followed by the robot is generated using the A∗ path connecting the selected points (as shown in Figure 13b). In trial-2.f, the point selection is done based on the PSO algorithm and in trial-2.d (Figure 13b), the point selection is done based on the ACO algorithm (Figure 13b). From trial-2.e to trial-2.h, we could observe that the optimal path generated by the ACO algorithm has a shorter path length than all other methods. The most sub-optimal strategy was a random selection of points. In the case of the ACO algorithm, the robot took 642 s to complete the exploration with a total energy consumption of 32.35 kJ. In environment-2, ACO shows the best convergence in optimizing the path length and energy consumption. Another important observation is zig-zag selection (trial-2.f) has outperformed PSO (trial-2.g) in yielding a shorter path length and less energy consumption. Similar to environment-1, the random selection of points recorded the most energy-expensive path generation strategy.

Environment-3 is the biggest area among all other environments where the algorithm has identified 59 sample points from 124 m${}^{2}$ of area. The trials conducted in environment-3 is represented as trial-2.i, trial-2.j, trial-2.k and trial-2.l respectively. In trial-2.i, the robot did sample auditing in the identified locations by selecting the points randomly and following an A∗ path connecting the selected points (shown in Figure 14a). In trial-2.j, the robot did sample auditing in the identified locations by selecting the points in a zig-zag fashion along the y-axis. Similar to the previous trial, the robot followed an A∗ path connecting the selected points (shown in Figure 14b). In trial-2.k, the point selection is made based on the PSO algorithm and in trial-2.d, the point selection is made based on the ACO algorithm. From trial-2.i to trial-2.l, we could observe that the optimal path generated by the ACO algorithm has a shorter path length compared to all other methods. The most sub-optimal strategy was the random selection of points. The robot took 951 s to complete the path generation in the case of the ACO algorithm with a total energy consumption of 47.89 kJ. From the observations made from environment-3, it is evident that ACO shows the best convergence in optimizing the path length and energy consumption.

From the experiments conducted in environment-1, environment-2 and environment-3, ACO showed a better performance in yielding shorter and energy-optimized paths for sample auditing. After ACO, the second-best path generation in terms of shorter path length and less energy consumption is given by PSO for environment-1 and Zig-Zag for environment-2 and environment-3. In larger environments, the PSO algorithm was showing a sub-optimal performance. The robot skipped few sampling points during the audit process since navigation algorithms in BELUGA robot was not allowing the robot to visit the narrow location. However, skipping a few samples (2 out of 39) while auditing has a negligible effect on the overall auditing process. Besides, the compact dimension of the sample-audit sensor allows using compact robot platforms to perform auditing in narrow space by trading-off requirements for large power sources and computation modules.

## 7. Conclusions and Future Works

A novel path planning strategy for robot-aided cleaning auditing has been devised by extracting the geometrical features from the map. Considering the boundaries and corners as the geometrical signatures that contributes to the dirt accumulation in an indoor environment, the locations for performing the auditing process are identified as part of the path planning strategy. To generate an optimal path that covers the identified sample locations, swarm algorithms like ACO and PSO are utilized. The optimization algorithm identified an efficient path covering all the sampling locations by minimizing the energy consumption by the robot. The dirt gathering efficiency of formulated geometry-based sampling locations and the behavior of the paths generated by the proposed approach are evaluated in real-time. Experiment results show that the geometry feature-based sample location identification aligned with dirt accumulation spots after multiple cleaning iterations in the same environment. The ACO-based path generation showed better performance by yielding the shortest exploration path with the smallest energy footprint compared to PSO and other path generation strategies like zig-zag and random point selection in our in-house developed BELUGA robot.

The future works of this research will be focusing on:Study the effect of variation of dirt patterns in auditing algorithms and consider dirt pattern distribution for audit path planningA comprehensive dirt dataset generation for machine learning-based sample auditingInclusion of more geometrical signatures that contribute to the dirt accumulation in a regionExtending the present cleaning auditing framework by including olfactory sensing techniques

## Figures and Tables

**Figure 1 sensors-22-05317-f001:**
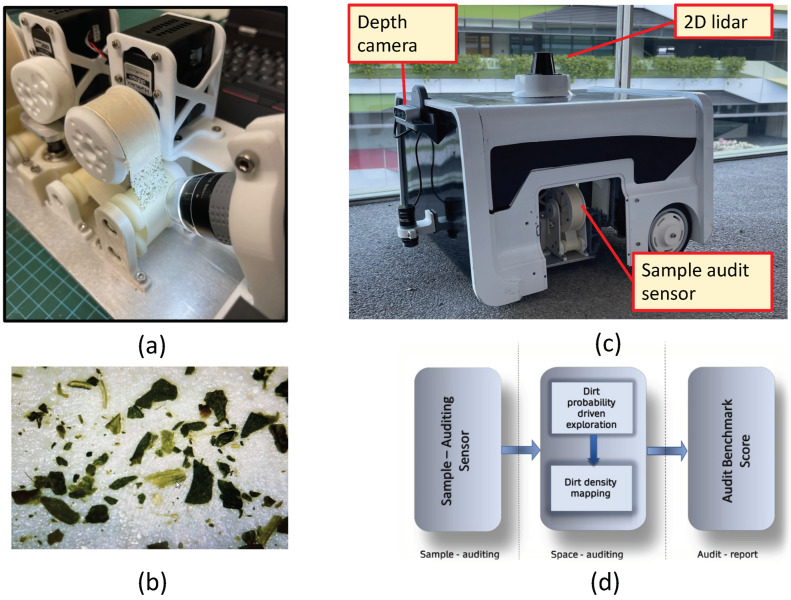
The dust extracted by the sample audit sensor (**a**), the dust particles viewed by the microscope of the sample audit sensor (**b**), the BELUGA audit robot equipped with sample audit sensor (**c**), and the overview of robot-aided cleaning auditing framework [19] (**d**).

**Figure 2 sensors-22-05317-f002:**
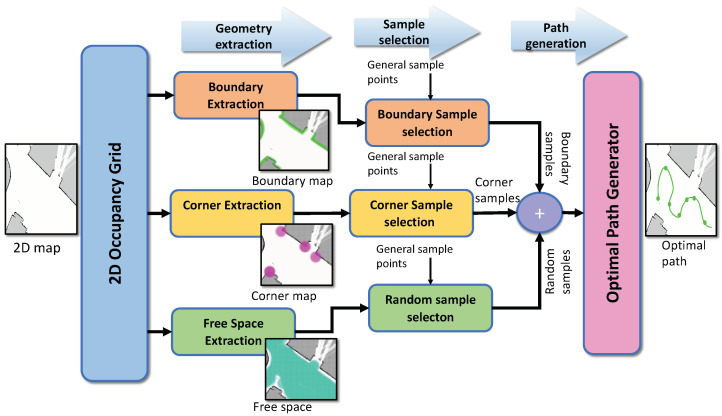
The overview of the proposed path generation strategy for dirt sample gathering using geometrical signatures extracted from the 2D map of the environment combined with an optimal path generator.

**Figure 3 sensors-22-05317-f003:**
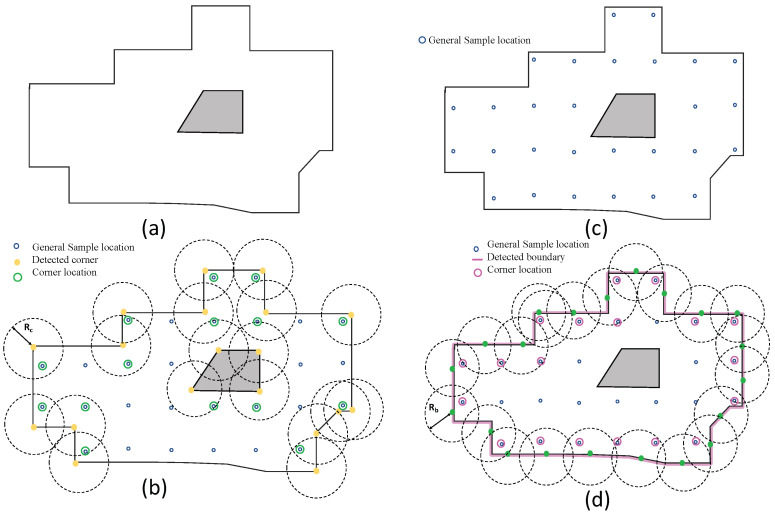
The representation of occupancy grid (**a**), corner location extraction (**b**), general sample locations after uniform sampling (**c**), and boundary location extraction (**d**).

**Figure 4 sensors-22-05317-f004:**
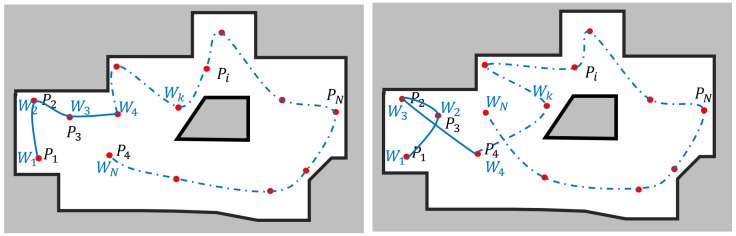
Examples for waypoint sequences that can be followed by the robot for auditing process.

**Figure 5 sensors-22-05317-f005:**
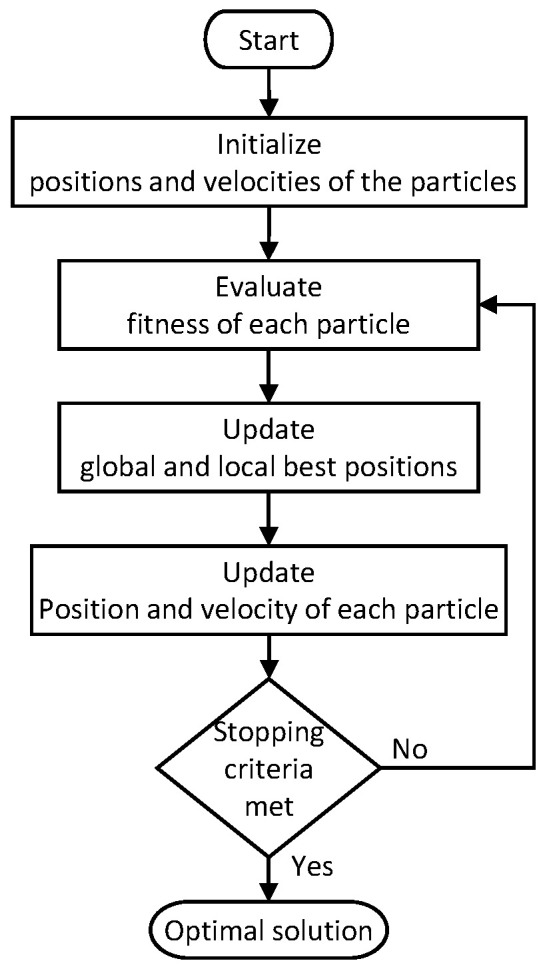
Flow of the PSO algorithm.

**Figure 6 sensors-22-05317-f006:**
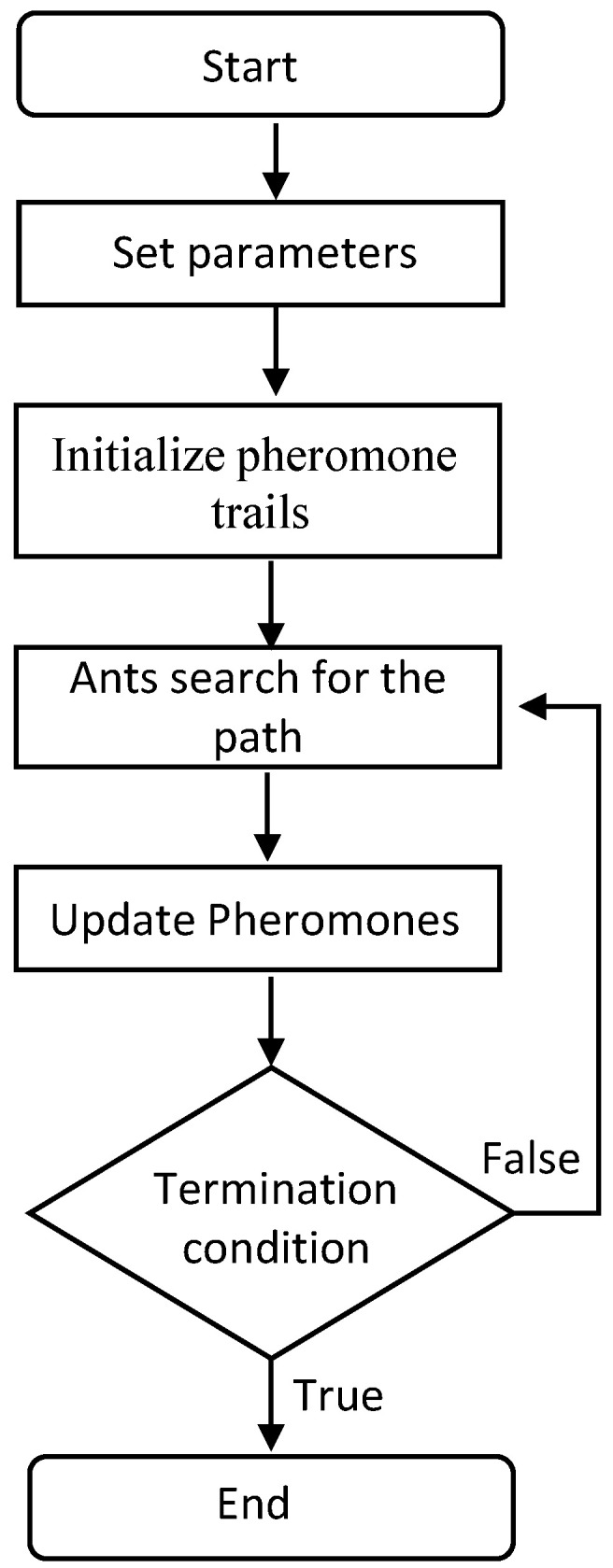
Flow of the ACO algorithm.

**Figure 7 sensors-22-05317-f007:**
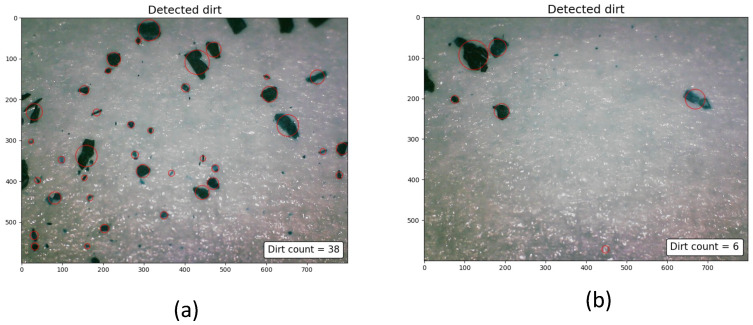
The dirt particles gathered from a dense dirt accumulated region (**a**), the dirt particles gathered from a sparse dirt accumulated region (**b**).

**Figure 8 sensors-22-05317-f008:**
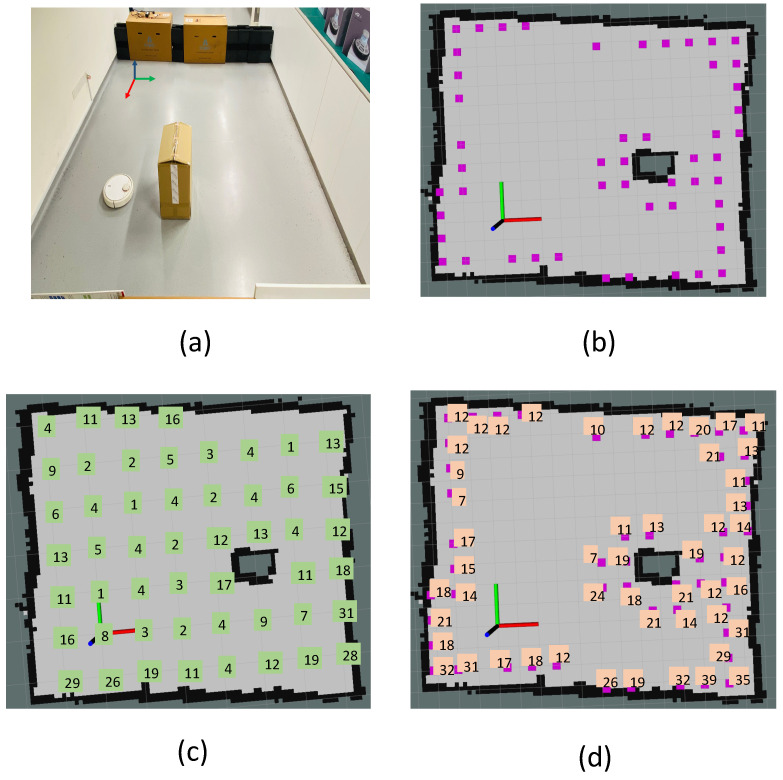
The environment chosen for trial-1.a (**a**), identified sample locations based on the proposed approach (**b**), dirt counts recorded corresponding to locations in a uniform grid sampling (**c**), dirt counts recorded corresponding to identified sample locations based on the proposed approach (**d**).

**Figure 9 sensors-22-05317-f009:**
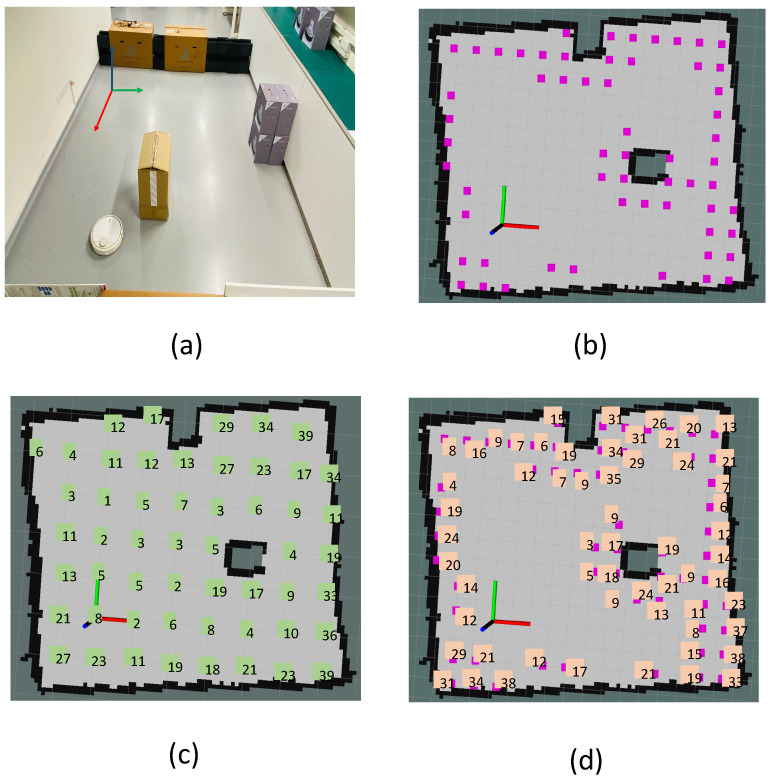
The environment chosen for trial-1.b (**a**), identified sample locations based on the proposed approach (**b**), dirt counts recorded corresponding to locations in a uniform grid sampling (**c**), dirt counts recorded corresponding to identified sample locations based on the proposed approach (**d**).

**Figure 10 sensors-22-05317-f010:**
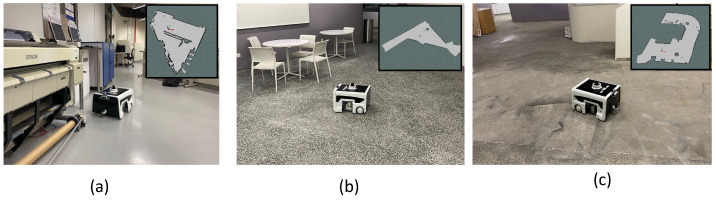
BELUGA robot operating in different environments (map given in inset). Environment-1 (**a**), environment-2 (**b**), and environment-3 (**c**).

**Figure 11 sensors-22-05317-f011:**
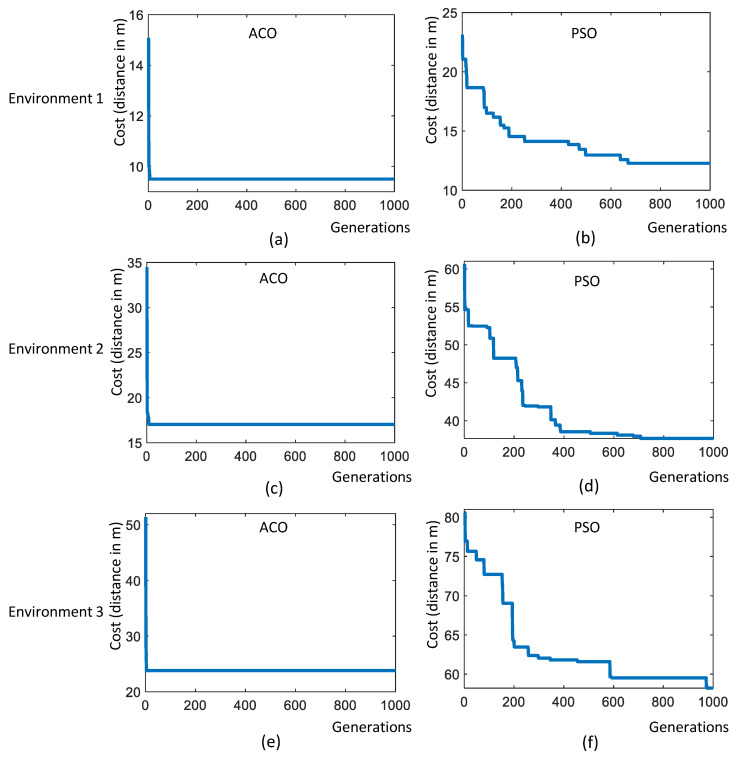
Convergence results of ACO and PSO. (**a**): Environment 1 using ACO, (**b**): Environment 1 using PSO, (**c**): Environment 2 using ACO, (**d**): Environment 2 using PSO, (**e**): Environment 3 using ACO, and (**f**): Environment 3 using PSO.

**Figure 12 sensors-22-05317-f012:**
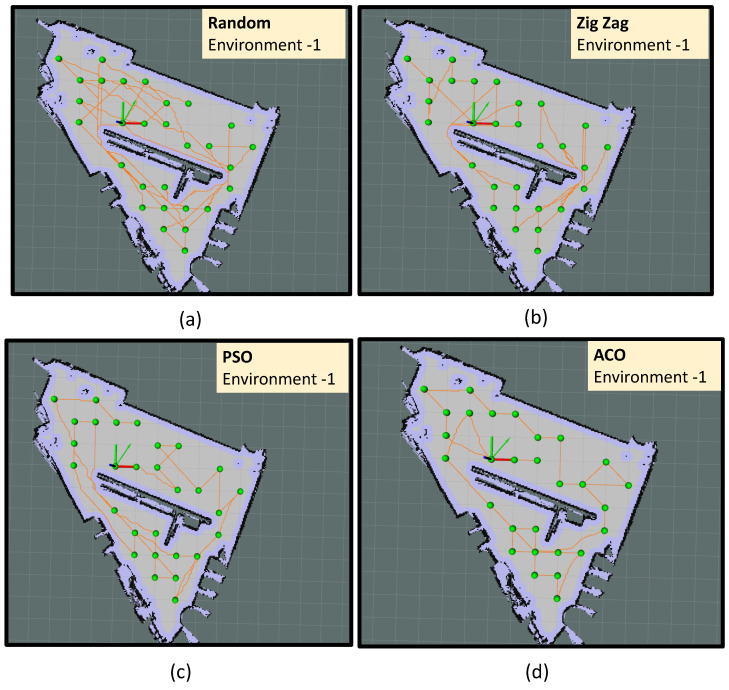
The path followed by the robot in environment-1 using random selection of points (**a**), using zig-zag selection of points (**b**), PSO algorithm (**c**), and ACO algorithm (**d**).

**Figure 13 sensors-22-05317-f013:**
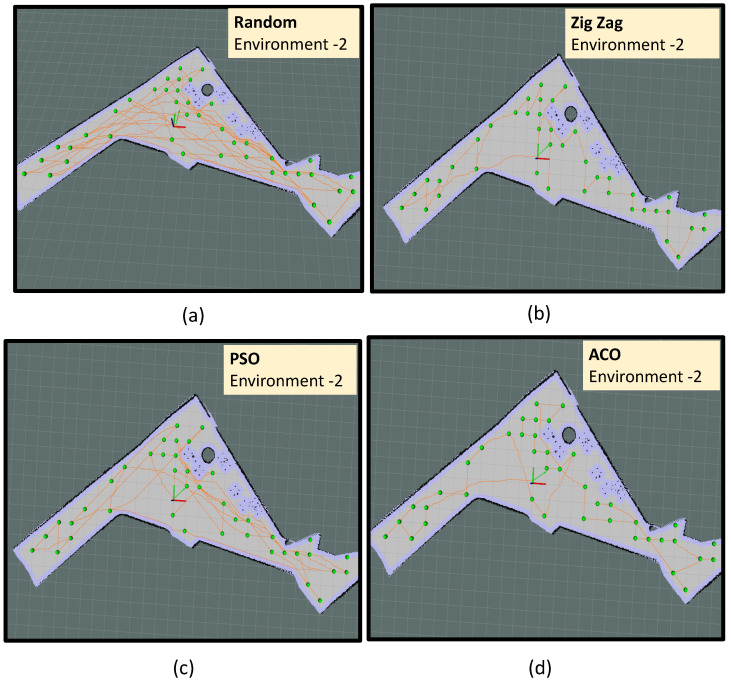
The path followed by the robot in environment-2 using random selection of points (**a**), using zig-zag selection of points (**b**), PSO algorithm (**c**) and ACO algorithm (**d**).

**Figure 14 sensors-22-05317-f014:**
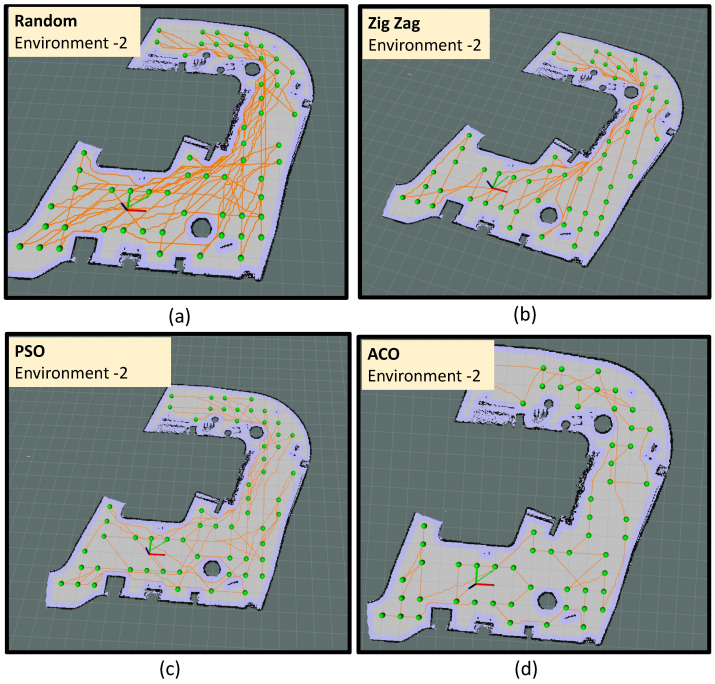
The path followed by the robot in environment-3 using random selection of points (**a**), using zig-zag selection of points (**b**), PSO algorithm (**c**) and ACO algorithm (**d**).

**Table 1 sensors-22-05317-t001:** The sampling efficiency for a dirt count threshold 10 in trial-1a.

	Trial-1a	
	Uniform Sampling	Proposed Approach
Number of dirt samples $N_{d}$	22	47
Number of clean samples $N_{s}$	29	4
Total sampling $\left(N_{d}+N_{s}\right)$	51	51
Dirt gathering efficiency $N_{d}/\left(N_{d}+N_{s}\right)$	0.43	0.92

**Table 2 sensors-22-05317-t002:** The sampling efficiency for a dirt count threshold 10 in trial-1b.

	Trial-1b	
	Uniform Sampling	Proposed Approach
Number of dirt samples $N_{d}$	29	44
Number of clean samples $N_{s}$	24	15
Total sampling $\left(N_{d}+N_{s}\right)$	53	59
Dirt gathering efficiency $N_{d}/\left(N_{d}+N_{s}\right)$	0.54	0.74

**Table 3 sensors-22-05317-t003:** The overall observations from trial-2.

Algorithm	Parameter	Validation Trials		
		Environment-1 N = 29	Environment-2 N = 39	Environment-3 N = 59
PSO	Path length (*D*)	12.27 m	37.66 m	58.21 m
	Time taken (*T*)	470 s	889 s	1320 s
	Total energy (*E*)	23.72 kJ	44.82 kJ	66.84 KJ
ACO	Path length (*D*)	9.5 m	17.04 m	23.8 m
	Time taken (*T*)	427 s	642 s	951 s
	Total energy (*E*)	22.55 kJ	32.35 kJ	47.89 KJ
Zig-Zag	Path length (*D*)	16.33 m	17.85 m	53.42 m
	Time taken (*T*)	511 s	665 s	1248 s
	Total energy (*E*)	27.98 kJ	35.13 kJ	65.88 KJ
Random	Path length (*D*)	25.15 m	73.31 m	106.75 m
	Time taken (*T*)	602 s	1220 s	1785 s
	Total energy (*E*)	31.78 kJ	64.44 kJ	94.39 KJ

## Data Availability

Not applicable.

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
