# Peer review of "A Novel Path Planning Strategy for a Cleaning Audit Robot Using Geometrical Features and Swarm Algorithms"

_sensors, 2022, doi:10.3390/s22145317_

Round 1

Reviewer 1 Report

The manuscript is well written and does not lack clarity. Please find my comments on this work below:

1) It would be useful if the authors could provide a metric that would compare the number of detected dirt samples to the number of available dirt samples.

2) How will the robot's performance change depending on how dirt is dispersed? I will encourage authors to carry out further experiments to investigate this.

3)  How would the performance alter if the environment changed, such as a robot walking on a different surface or in a different light environment?

4) Please describe the simulated environment in further detail.

5) Please ensure that in et al. et is not followed by a period. There should be a period after al. Please also ensure that everything is in lowercase letters.

Reviewer 2 Report

This paper is motivated by the use of autonomous robots to assess the cleanliness level of a given region, by automatically analysing the dirt samples collected from several locations of that region. Overall, the paper is well organized and experimental results are provided to validate the proposed approach in a real-world environment, using an in-house developed cleaning auditing robot BELUGA.

The main contributions of this work seem to be: (1) a heuristic approach to identifying the best location candidates to perform a dirt level assessment; (2) an alleged optimal path planning strategy to visit all the candidate locations, that minimizes the energy the robot takes to visit all these locations. However, there seems to be some strong scientific flaws regarding both these contributions:
 1 - Identification of location candidates for dirt level assessment: the proposed heuristic approach seems reasonable and sensible arguments are provided to justify the sampling of candidate locations near corners and walls. However, the experimental validation is not sound, as the authors simply do not provide a quantitative metric to evaluate the proposed approach. As it is, the problem formulation is very vague: what exactly are the authors trying to achieve? What is the problem ground truth? For instance, for typical dirt distributions in a given region, after cleaning is performed by an autonomous cleaning robot, is this method supposed to minimize the error between true cleanliness level, averaged over all the region, and estimated average cleanliness level, after sampling the proposed locations? How does one quantify the proposed approach performance? What is the robot used for cleaning the region (there is only a mention to a "commercially available domestic floor cleaning robot with visual-inertial navigation capabilities")? What is the region cover algorithm this robot is running? Will a different cleaning robot generate a different remaining dirt pattern in the floor? If the authors want to evaluate location candidates for cleanliness auditing this is vital information! Also, some cleaning robots may leave some dirt in some locations due to the difficulty in reaching such locations. Given the geometry of the BELUGA robot, is it reasonable to expect this audit robot to reach such locations?
 The authors only provide some qualitative results (Figure 7) but these are not convincing at all, as they seem the result of some "cherry picking" approach. I also don't understand why there are not candidate points for dirt level assessment along some walls in trials 1a and 1b (northwest wall, Fig. 7a, and southwest wall, Fig. 7d). According to the proposed method and Fig. 3d, there should be more sampled boundary locations along these walls. (BTW, in Fig. 7a and 7d there are purple and light blue dots in the map: what is their meaning?)
 2 - Path generation: the authors make bold claims regarding the "optimality" of the proposed path optimization algorithm: "This combined approach generates an optimal path covering all the identified dirt locations" (line 7); "The optimal path generator in the proposed framework takes in the locations identified after the sample selection procedure and generates an optimal path with minimum time and energy" (line 202-204); "Swarm optimization algorithms are effective techniques for finding the optimal solution for this kind of problem" (lines 282-283); "The optimization algorithm identified the best path covering all the sampling locations by minimizing the energy consumption by the robot." (lines 440-442). As the authors acknowledge, this optimization is an NP hard problem, and the proposed PSO and ACO only provide an approximation to the best solution, but they do not guarantee its optimality (as they run a limited number of iterations). The path generation problem addressed in this paper is of course the classic Travelling Salesman Problem, although the authors do not identify it as so. There is an extensive literature regarding this problem, and many efficient approaches besides PSO and ACO that can provide optimal results for a moderate number of points to visit (e.g., branch-and-bound algorithms, progressive improvement algorithms, etc.), and other approximation algorithms like the multi-fragment algorithm and greedy algorithms. So I really don't see the point of experimentally evaluating these two algorithms (PSO and ACO) against random travelling and the "Zig-Zag" method (which, BTW, is not clearly stated, the definition "selecting the points in a zig-zag fashion along the y-axis" is not clear), as this is a classical problem for which there are a lot of better algorithms out there and for which extensive theoretical results exist.

Bibliography: Some references seem incomplete (e.g., 2, 21, but check all other references). Also, regarding path planning for autonomous cleaning robots, references 21-23 hardly seem to be the state-of-the-art (lines 91-94).

English level: the paper needs a substantial editing of English language and style. Many expressions used in the paper are strange and there are many typos. Some examples (this is not by all means an extensive list):
    10: "device" instead of "devised";
    24: "fruitful invasion";
    60: etal.
    63: et.al
    71: The founding effort
    260: "However, the number of random locations are smaller in number and it is selected based on the size of the occupancy grid" (bad phrasing, no final punctuation mark)
    262: "The locations for performing sample auditing is the combination of corner locations, boundary locations and random locations, as represented in Eq 1" (idem)
    277: "becomes N(N − 1)/2, which is higher if N is high" ?
    319: "Our experiment trials comprised of"
    367: "where the environment is more cluttered and *disoriented* in terms of obstacles"
    377: "The the"
    And so on...

Round 2

Reviewer 1 Report

Thank you so much for the revised manuscript. I am really pleased with the changes made.

Please check that reference # 58 (particularly the authors' names), in the reference section, is appropriately written. Instead of writing an et al., please list all of the authors' names in the reference section.

Author Response

Respected reviewer,

 We sincerely thank you for reviewing our manuscript and providing valuable suggestions for improvement. We considered your comments seriously and carefully revised our manuscript.

The reply to the comment is given below:

Thank you so much !

Sincerely,

Thejus Pathmakumar

Comment -1

Please check that reference # 58 (particularly the authors' names), in the reference section, is appropriately written. Instead of writing an et al., please list all of the authors' names in the reference section.

Reply

Thank you so much for suggesting the correction. Sincere apologies for the formatting error in the reference section. We have replaced the et al. usage with the name of authors.

The changes are made in the reference section.

Reviewer 2 Report

Overall, the authors have made a significant effort to revise the paper and to address the questions and problems stated in the previous revision. However, there are still some point of concern that should be addressed in the final version of the paper:
    - I have some problems regarding the proposed metric (line numbers: 327 - 394):
        1) The authors state that their "first set of experiments is designed to analyze the ability of the proposed approach to identify dirt accumulated locations and not to analyze the cleanliness level". However, later they say that "the overall objective of the cleaning auditing is to estimate how clean the location is based on the samples collected. To perform auditing the cleanliness, it is not necessary to know the exact dirt pattern left by the robot as long as the overall audit score is proportional to the cleaning quality delivered by the robot". There is a problem with this approach: if the method is searching for dirt accumulated locations, later, when used for cleaning auditing, it will generate a strong bias towards a high dirt level, as it will tend to skip areas considered cleaned by the algorithm. Consider this example: to estimate the male/female proportion on a given population, an algorithm picks samples of persons having a beard. Do you think this is a good approach?
        2)  The metric you are using (eq. 4) is the *precision*. However you should use the *recall*, as the precision does not consider the ground truth at all and as such it does not convey much meaning. One more example: there is some dirt on the right side of a room, but substantially more dirt on the left side of the same room. If the robot chooses to inspect, say, only 5 location points on the right side of the room, it can eventually gather 5 dirt samples and thus achieve a 100% score (according to eq. 4), even if it did not manage to inspect the dirtiest regions of the room. Is this reasonable? Of course, it can be difficult to obtain the recall (ratio of dirt samples identified by the dirt location identification algorithm to the true number of dirt locations in the room), as this is a kind of ill-posed problem, but the authors should at least perform a clear and rigorous analysis of this evaluation problem.
    - Regarding the path generation:
        1) There is no "approximated optimal path": a path is optimal or not. (Lines 8, 110, 115, 205, 284, 491, etc). I think you should simply use "efficient path planning" or something like this;
        2) "However, in the classical TSP, it is assumed that that there are no obstacles in between the nodes and hence Euclidean distances between the nodes are considered for the optimization." This is a false statement. TSP deals with optimizating a path on a graph, where the edges connecting its nodes are associated with a given cost. You can use any cost you want between nodes (locations to visit): Euclidian distance, A* distance, energy needed to travel between nodes, time, etc. Please reformulate lines 290-297.
        3) "In that case, the number of points will be much higher, and optimization methods would be more suitable than the other approaches". I really don't understand the point of the authors here: this is an optimization problem, and so *all* the approaches are optimization methods.
        4) Since the total number of locations to visit is less than 60 in all 3 experiments, the authors should provide the *optimal* path according to the A* distances between every pair of nodes (there should probably exist a lot of open-source code available out there to obtain this optimal solution), and then compare the solutions they provide to this baseline, in order to assess the efficiency of the proposed methods. This would greatly improve the paper.
    - There are some typos and many grammar errors in the new text (in red in the document). In particular, pay attention to singular/plural agreement (for instance, have/has error appears many times).
